# Progress in Bioengineering Strategies for Heart Regenerative Medicine

**DOI:** 10.3390/ijms23073482

**Published:** 2022-03-23

**Authors:** Timm Häneke, Makoto Sahara

**Affiliations:** 1Department of Cell and Molecular Biology, Karolinska Institutet, 171 77 Stockholm, Sweden; timm.haneke@stud.ki.se; 2Department of Surgery, Yale University School of Medicine, New Haven, CN 06510, USA

**Keywords:** biomaterial, biofabrication, bioprinting, cardiomyocyte maturation, cardiac tissue engineering, cell therapy, heart disease, regenerative medicine, stem cell biology

## Abstract

The human heart has the least regenerative capabilities among tissues and organs, and heart disease continues to be a leading cause of mortality in the industrialized world with insufficient therapeutic options and poor prognosis. Therefore, developing new therapeutic strategies for heart regeneration is a major goal in modern cardiac biology and medicine. Recent advances in stem cell biology and biotechnologies such as human pluripotent stem cells (hPSCs) and cardiac tissue engineering hold great promise for opening novel paths to heart regeneration and repair for heart disease, although these areas are still in their infancy. In this review, we summarize and discuss the recent progress in cardiac tissue engineering strategies, highlighting stem cell engineering and cardiomyocyte maturation, development of novel functional biomaterials and biofabrication tools, and their therapeutic applications involving drug discovery, disease modeling, and regenerative medicine for heart disease.

## 1. Introduction

Heart disease is a leading cause of mortality worldwide and still continues to be subject to poor prognosis, irrespective of development in the current diagnosis and treatment strategies involving pharmacological therapy and revascularization interventions [1,2]. Among tissues and organs, the mammalian hearts have the least regenerative capabilities, most of which are lost after 10 years of age in humans [3]. Given the drastic shortage of donor hearts for transplantation, which is only the radical treatment for severe heart failure (HF), developing new therapeutic strategies for severe forms of heart disease is urgently required but remains elusive in modern cardiac biology and medicine [4]. In this regard, recent advances in stem cell biology and biotechnologies, such as human pluripotent stem cells (hPSCs) and cardiac tissue engineering, hold great promise for opening novel therapeutic paths to heart regeneration and repair for heart disease [5,6]. To date, however, the therapeutic effects by the cell-based therapies in the preclinical and/or clinical settings remain insufficient with only modest outcomes, likely due to the reason that the benefits by cell therapies are mainly derived from the paracrine factors that are released by transplanted cells but not from direct cell engraftment replacing damaged heart tissues. A majority of cells (≈95–99%) that are delivered to the injured hearts are lost within the first 24 h, owing to mechanical washout, death due to a hostile environment (i.e., poor oxygenation and low nutrients), and host immune reaction [7,8,9,10]. On the contrary, more recent studies have revealed that combining the cell-based therapy with bioengineering technology, which is represented by functional biomaterials and biofabrication tools, can promote the engraftment of transplanted cells, thereby leading to improved therapeutic effects, although these areas are still in their infancy [11,12]. In this review, we summarize the latest progress in cardiac tissue engineering strategies, highlighting stem cell engineering and cardiomyocyte maturation, development of novel functional biomaterials (e.g., hydrogels and decellularized scaffolds) and biofabrication tools (e.g., engineered heart tissues and bioprinting), and their therapeutic applications involving drug discovery, disease modeling, and regenerative medicine for heart disease. Finally, we discuss the current challenges and future perspectives of the cardiac tissue engineering technologies from the viewpoints of their clinical significance.

## 2. Stem Cell Engineering and Cardiomyocyte Maturation

### 2.1. Cell Sources for Cardiac Bioengineering

Both human embryonic stem cells (hESCs) and human induced pluripotent stem cells (hiPSCs), namely hPSCs, have their high self-renewal capacity and capability to differentiate into any cell type of the human body [13,14]. Using the established protocols to differentiate hPSCs highly efficiently into cardiomyocytes (CMs) [15,16], vascular endothelial cells (ECs) [17,18], and vascular smooth muscle cells (SMCs) [19,20], these cardiac cells forming the human heart can be generated and obtained almost unlimitedly in vitro. Therefore, exploiting hPSCs is an attractive approach for cardiac bioengineering, disease modeling (e.g., using patient-derived hiPSCs), and regenerative medicine [6,21,22,23].

Sophisticated differentiation protocols with modulation of the Wnt signaling pathway can generate hPSC-derived CMs (hPSC-CMs) with >90% purity and scaling up to 10^9^ cells [15,16,24,25,26]. However, these hPSC-CMs are structurally, functionally, and transcriptionally more similar to fetal rather than adult CMs [27,28,29], which hampers the broad applications of these in vitro-generated CMs in the various settings. For instance, the electrical immaturity of hPSC-CMs might cause ventricular arrhythmias that appeared following cellular engraftment after cell transplantation in animal injured hearts [30,31]. Therefore, a wide variety of the maturation strategies for hPSC-CMs have been investigated and employed [32], as described in the following section. Although the immature characteristics of hPSC-CMs are usually considered as their major demerits, some properties of the immature hPSC-CMs might be still advantageous in a specific application, such as cell transplantation after cardiac injury for therapeutic regeneration. Reinecke et al. have reported that fetal and neonatal CMs could engraft in normal and injured myocardium in rat myocardial infarction (MI) models, while adult CMs could not survive nor engraft upon transplantation into the host myocardium [33], suggesting that fully mature CMs might not be suitable for the cell therapy application. Taken together, it is essential to determine the optimal state of cellular maturity for hPSC-CMs at the timing of cell transplantation into the damaged hearts, in order to obtain the best results for cell engraftment and heart regeneration.

The recent advanced studies such as single-cell/nuclei RNA sequencing analyses have revealed a cellular heterogeneity and molecular signatures in human fetal and adult hearts [34,35,36,37]. Notably, the cellular composition, including CMs, ECs, SMCs, fibroblasts and immune cells, and their transcriptional profiles are different depending on each of the anatomical regions across the heart, suggesting differential roles and functions of the various heart regions, such as cardiac outflow tract, atria, and the ventricular septum and free wall [36,37]. This complex heterogeneity in the in vivo heart cells may affect the future strategies in cardiac tissue engineering upon the selection and employment of the ideal cell populations among hPSC-CMs.

### 2.2. Cardiomyocyte Maturation Strategies

As noted above, a major obstacle for the use of hPSC-CMs in various applications is their immature characteristics with respect to their structural morphology (i.e., lacking a rod-like cell shape, less aligned sarcomeres, undeveloped myofibrils and T-tubules, no/less polyploidy or binucleation, no/less polarized intercalated discs, and a low density of mitochondria); contractile force (e.g., their negative force-frequency relationship [FFR]); metabolism (mainly employing anaerobic glycolysis, but not β-oxidation of fatty acids, as a major source of ATP production); cell cycle (active cycling and division); electrophysiology (e.g., less negative resting membrane potential, slower upstroke velocity, and a faster repolarization phase without a plateau phase); calcium handling (i.e., slower action potential propagation due to inefficient excitation-concentration coupling); and transcriptomic levels [27,28,29,32,38,39]. To overcome these immaturity issues and promote maturation in hPSC-CMs, various bioengineering strategies have been investigated and employed to date, and several effective approaches are briefly described below and listed in Table 1. These strategies involve mechanical loading and electrical stimulation, three-dimensional (3D) cultures including engineered heart tissues (EHTs) with a scaffold protein (e.g., a hydrogel, an extracellular matrix [ECM], etc.) or cardiac cellular aggregates (spheroids) without scaffold proteins, intercellular crosstalk via co-culturing with non-cardiac cells, metabolic and hormonal interventions, genetic regulation, and in vivo transplantation (Table 1).

hPSC-CMs are responsive to external stimuli such as mechanical force and electrical pacing, and in fact, these mechanical and electrical stimuli are demonstrated to promote functional and structural maturation of hPSC-CMs in vitro, respectively [40,41,44,45]. Further, a recent study has proven that hPSC-CMs, which were electrically- and mechanically-stimulated simultaneously in a combination with cyclic stretch and electrical pacing (from 2 to 6 Hz) for two weeks, resulted in the more matured CMs with adult-like characteristics, such as a broad network of T-tubules, a positive FFR, and enhanced responses to a β-adrenergic stimulator [42]. Nowadays, 3D culture systems, including EHTs and cardiac spheroids/organoids, are recognized as one of the most successful approaches to promote the maturation of hPSC-CMs in vitro [46,47,48,49], as described in the following sections. The presence of non-cardiac cells, such as vascular ECs and SMCs and fibroblasts (including epicardial cells) also promotes CM maturation via intercellular crosstalk and paracrine signals [55,56,57,58]. The co-culture and/or co-administration of hPSC-CMs with ECs, SMCs, and/or fibroblasts resulted in the more developed muscle constructs with enhanced CM maturity and improved an efficacy of CM engraftment upon transplantation into the host myocardium, compared to hPSC-CMs alone [52,53,54]. One of the main characteristics for CM maturation in vivo is a metabolic shift from glycolysis to β-oxidation of fatty acids, mirroring a morphological change to cellular hypertrophy and increased energetic demands [66]. In line with this notion, the modified culture medium with high fatty-acids and low glucose contents was proven to drive metabolic, functional, and structural maturation of hPSC-CMs in vitro [59,60]. Among the hormones whose levels in the body peak around the time of birth, a thyroid hormone tri-iodothyronine (T3) and a glucocorticoid hormone (e.g., dexamethasone) have been shown to improve the maturation of hPSC-CMs in vitro, respectively. When these two hormones were simultaneously applied to the cultured hPSC-CMs, they exerted further maturation of the CMs in a synergistic manner [61,62,63].

For a more in-depth review about the various strategies for cardiac maturation, including other approaches, we direct the readers to other reviews on the topic [23,32,65].

## 3. Development of Functional Biomaterials for Cardiac Tissue Engineering

Biomaterial-based strategies for tissue engineering have recently attracted much attention in cardiac biology and medicine, since they are known to be effective for not only promoting the maturation of hPSC-CMs in vitro but also improving the cardiac regenerative effects of cell-based and/or cell-free therapies against heart disease models in vivo [11,12]. Biomaterials, represented by hydrogels and decellularized scaffolds, have various advantages for regenerative therapies, including enhancing cell survival and cell retention/engraftment into the host tissue; promoting protective paracrine signaling responses by containing and releasing functional bioactive molecules such as angiogenic, anti-apoptotic, and/or immunomodulatory factors; and providing mechanical support for the damaged tissues [64,67,68]. Ideal biomaterials in tissue engineering are usually considered as being biocompatible, biodegradable, less immunogenic, and biomimetic products resembling properties of the target native tissue.

### 3.1. Hydrogels

Hydrogels are the most popular and widely used polymers among the various bioengineering materials [69,70]. They can absorb a high amount of water and biofluids, similar to human tissue, and be swollen maintaining their shapes until being degraded by hydrolysis [70,71]. Hydrogels also have several beneficial properties for the use in tissue engineering as follows: injectable fluid in a less invasive fashion [72], efficient transfer of oxygen and metabolites [73,74], ability to encapsulate cells and bioactive molecules [75], and providing casting molds of several geometries [40,76] and anchoring/mechanical support [42,77]. Three types of hydrogels, i.e., ones derived from either natural, synthetic, or hybrid biomaterials are known and used in a wide variety of applications in tissue engineering. Natural hydrogels include fibrin, collagen, gelatin, alginate, chitosan, hyaluronic acid, and elastin [78,79,80,81,82,83], while synthetic hydrogels include poly(ethylene glycol) (PEG), poly(lactic-*co*-glycolic acid) (PLGA), polycaprolactone (PCL), poly(N-isopropylacrylamide) (PNIPAAm), etc. [84,85,86,87]. Recently, hybrid hydrogels that contain the natural and synthetic biomaterials combined with covalent binding or crosslinking have been developed as an alternative option [88,89].

Early studies reported that acellular injectable hydrogels alone provided efficient mechanical support to the infarcted heart muscles and elicited therapeutic effects through attenuating pathological ventricular remodeling in small and large animal models [69,90,91,92]. Fibrin, a biodegradable coagulation-related protein, is a natural biomaterial that has been broadly used in cardiac tissue engineering [82,93,94,95]. Christman et al. demonstrated the therapeutic effects of fibrin glue as an acellular injectable biomaterial in rat MI models via mechanical support and induction of angiogenesis by recruiting ECs [94,96]. Fibrin gels/patches are also manufactured frequently encapsulating hPSC-CMs with or without other cell types, such as ECs and/or SMCs. For instance, the cardiac patches created from hiPSC-CMs and human pericytes entrapped in a fibrin gel reduced infarct sizes and improved cardiac function in a rat MI model, together with survived and proliferated human CMs and pericytes upon transplantation [95]. Another group reported that transplantation of insulin-like growth factor (IGF)-loaded 3D fibrin patches combined with hiPSC-CMs, hiPSC-ECs, and hiPSC-SMCs resulted in significant improvements in left ventricular function, infarct size, myocardial wall stress, and myocardial hypertrophy without inducing malignant arrhythmias in a porcine MI model [82,97]. Interestingly, in their system, the cell engraftment rate was relatively high (11 ± 2%) at four weeks after the transplantation, indicating the strong potential of the fibrin gels to promote transplanted cells’ survival and retention into the host environment.

Collagen is a natural fibrous protein that is abundant in the ECM and is also used widely in tissue engineering [81,98,99]. Engineered heart muscles that were generated by casting hESC-CMs with collagen in preformed molds successfully improved regional ventricular wall movement and preserved left ventricular (LV) systolic function upon transplantation into damaged rat hearts one month after ischemia reperfusion injury, which was measured with cardiac tagged magnetic resonance imaging [99].

Alginate, a natural linear polysaccharide that is derived from seaweed algae, has recently attracted much interest due to its ECM-like properties and biocompatibility. In fact, it has been used in various applications for tissue engineering [100,101]. Intramyocardial or intracoronary injection of alginate hydrogels was shown to be a feasible, safe, and effective acellular strategy that attenuated adverse cardiac remodeling and ventricular dysfunction in recent and late MI models in rats and pigs [102,103]. Further, a commercially available calcium alginate hydrogel (Algisyl-LVR^TM^) was tested for its ability to reduce LV wall stress and improve LV function in HF patients (ejection fraction [EF] <40%) that were undergoing coronary artery bypass grafting (CABG) [91]. This first-in-man pilot study also showed the clinical benefits such as decreased end-systolic volume and increased LV wall thickness and EF three months after treatment, moving forward into the randomized clinical trials, as described below.

Chitosan is another natural polysaccharide and recently it is often used in tissue engineering, since it has biocompatible and biodegradable natures, as well as anti-microbial and pro-angiogenic properties [104,105]. Chitosan thermosensitive hydrogels are liquid at 25 °C but gel at 37 °C. Importantly, intramyocardial injection of chitosan hydrogels could efficiently increase LV myocardial wall thickness, decrease infarct size, attenuate pathological LV remodeling, and preserve LV contractility in a rat MI model [106].

Synthetic hydrogels have also been examined for their tissue engineering properties and therapeutic potential in the forms of acellular or cell-containing injectable materials [84,85,86,87]. PEG is the most often used synthetic polymer in tissue engineering, due to its lower immunogenicity. In fact, PEG was shown to elicit ventricular protective effects and its functional improvement when injected as an acellular injectable material in rodent MI models [90,107,108]. Further, PEG hydrogels containing hiPSC-CMs and/or erythropoietin attenuated LV remodeling and improved LV function in a rat MI model, although no engrafted cells were detected, implying that the observed beneficial results were derived from paracrine effects that were induced by the injected cell/biomolecule-encapsulating PEG hydrogels [85].

Micro-grooved thin and biodegradable PLGA film (≈30 μm in thickness) with hPSC-CMs can be used for the construction of a biomimetic cardiac patch. Chen et al. demonstrated that this biomimetic cardiac patch showed steady-state contraction, easily developed under regular electrical stimuli, and recapitulated the anisotropic electrophysiological feature of native cardiac tissues, indicating the remarkable biocompatibility of PLGA [109]. As proved by the reduced incidence of arrhythmia, the micro-grooved PLGA film-derived cardiac patch was much more refractory to premature stimuli than the non-grooved PLGA film-derived patch, highlighting the therapeutic potential of this system for injured hearts. Another group has reported that multilayered hiPSC-CMs that were cultured and organized on aligned nanofibers made of PLGA polymer successfully engrafted into the damaged host hearts and improved the LV function four weeks after treatment in a rat MI model [110].

PNIPAAm is a thermosensitive polymer that holds its liquid state at room temperature and forms a gel at human body temperature [111]. The intramyocardial injection of biodegradable PNIPAAm-based hydrogels reduced collagen deposition, preserved LV wall thickness, and increased neovascularization in the injured heart environment, leading to the improvement of LV function in rat MI models [87,112]. Further, Navaei et al. reported that a biohybrid thermosensitive PNIPAAm-gelatin-based hydrogel promoted the survival and maturation of the encapsulated neonatal rat CMs that were co-cultured with cardiac fibroblasts (2:1 ratio) in vitro, as demonstrated by their superior structural organization, beating behavior, and cell-cell coupling [89]. As such, this study highlighted the well-organized bioactivity and mechanical robustness of the PNIPAAm-gelatin hybrid hydrogel system for cardiac tissue engineering. 

Overall, various injectable hydrogels with or without containing cells to date have successfully treated the infarcted heart muscles by preventing pathological remodeling and preserving LV function in animal MI models. Further, because hydrogels can serve as a delivery vehicle to encapsulate the bioactive factors, the combination approach of injectable hydrogels with the delivery of biomolecules, such as angiogenic (e.g., vascular endothelial growth factor [VEGF], basic fibroblast growth factor [bFGF]) and anti-apoptotic/pro-proliferative (e.g., stromal cell-derived factor 1 [SDF-1], IGF-1, neuregulin-1β) factors holds great promise as an attractive strategy of the new bioengineering therapies [64,113]. Hydrogels can also be modified and functionalized with cell-responsive and cell-binding peptide sequences, such as the Arg-Gly-Asp (RGD), Arg-Gly-Asp-D-Phe-Lys (RGDFK), and Tyr-Ile-Gly-Ser-Arg (YIGSR) peptides, which are derived from ECM molecules including collagen, fibronectin, and laminin, and may thereby exert multiple bioactive responses [114,115,116,117].

Despite the benefits and promising results of the biomaterial-based hydrogel therapies for heart disease in preclinical models, a number of obstacles and limitations still hinder their clinical translation. One disadvantage of the biomaterials, in particular natural ones, is that they may retain some surface antigens to exert unignorable foreign body reaction in the host [64,118,119]. As controlling with this immune response is essential for successful functionality and sustainability of implanted hydrogels, there are a lot of attempts to develop biomaterials that can modulate these immunologic reactions, which involve functionalization with appropriate bioactive molecules of immunomodulatory factors (e.g., interleukin-10 [IL-10], tissue inhibitor of metalloproteinase-3 [TIMP-3]) [120,121,122]. Another concern when transplanting a hydrogel into the myocardial wall is its potential effect causing arrhythmias. In this regard, interestingly, Suarez et al. reported that a PEG-based highly spreading hydrogel with its slow gelation time did not cause any conduction abnormalities when it was injected into normal and infarcted rat hearts, while a hydrogel exhibiting minimal interstitial spread due to its quicker gelation time created a substrate for arrhythmia shortly after injection, indicating the site of delivery and interstitial spread characteristics as strong factors to predict possible arrhythmias after biomaterial therapies for damaged hearts [123]. This type of a validation study of biomaterials is important and needs to be repeated in large animal models for their further clinical translation. Other concerns around the use of biomaterial-based hydrogels include low availability of the required clinical-grade biomaterials, batch–batch variability, difficulty for sterilization and standardization operating procedures, and uncertainty of the optimal time window and delivery route. For a more in-depth review about this subject, we direct the readers to other reviews [67,124,125,126].

### 3.2. Decellularized Bioscaffolds

Another effective approach to generate bioscaffolds in cardiac tissue engineering is to obtain inherent ECM ultrastructure and composition directly from the native heart tissues by decellularizing and removing resident genetic materials in them [127,128]. Several protocols for decellularization are developed using physical, chemical, and/or enzymatic approaches, and among them, ionic and nonionic detergents such as sodium dodecyl sulfate (SDS) are commonly used for the decellularization of cardiac tissues [127,129,130]. In early studies, decellularized cardiac ECM hydrogels, that were specifically derived from porcine heart ventricles, were administered into infarcted rats and pigs via percutaneous trans-endocardial injections two weeks after MI [131,132]. These studies showed the decellularized ECM hydrogel therapy increased the endogenous CMs in the infarct areas and preserved LV function post-MI without inducing arrhythmias, highlighting the feasibility, safety, and efficacy of this novel treatment strategy for MI. Further, D’Amore et al. reported that a bi-layered polyurethane-decellularized ECM cardiac patch attenuated ischemic ventricular wall remodeling such as scar formation and LV wall thinning and dilation, along with promoting angiogenesis [133]. Interestingly, the ECM patch material promoted recruitment of the anti-inflammatory M2 macrophages into the ischemic border zones of the infarct heart muscles, rather than the proinflammatory M1 macrophages, possibly explaining the mechanism behind the observed beneficial effects in part. More recently, a decellularized porcine cardiac ECM scaffold was engineered with biodegradable PLGA microcarriers and human cardiac stromal cell-secreted factors [134]. This hybrid acellular cardiac patch maintained its properties after long-term cryopreservation and showed therapeutic regenerative effects by reducing scarring, promoting angiomyogenesis, and preserving LV function in rat and porcine MI models.

Compared to decellularization, i.e., removing all cellular materials in the native tissue, recellularization into the decellularized matrix remains challenging. In a recent study, for recellularization of a human heart, 500 million hiPSC-CMs were injected in a ≈5 cm^3^ volume of the LV free wall of a decellularized human heart [135]. This approach resulted in the survival of approximately half of the injected cells that formed an immature cardiac tissue in the matrix. Although the input cell density (100 million/cm^3^) was much higher than a normal CM density in the human heart (10–28 million/cm^3^) [3], these injected cells yielded only a small area of contracting tissue generating a pressure of <3 mm Hg [135], indicating the limitation of the current recellularization technology. Irrespectively, decellularized human cardiac ECM scaffolds also have another intractable issue, i.e., few opportunities to be used due to the shortage of donor organs.

## 4. Implementation of Biofabrication Tools for Cardiac Tissue Engineering

### 4.1. Microfabrication Technologies for Generating Engineered Heart Tissues

The hydrogel system (i.e., a water-swollen polymer network) is simple and the most used technology for generating EHTs, as noted above. Besides, a wide variety of other microfabrication technologies for EHTs, including cell sheets [136,137], muscle strips [40,43,45,76,138,139], cardiac patches [140,141], spheroids [50,51] and organoids [142,143,144], and electrospinning [145,146], have been developed and allowed more advanced cardiac bioengineering to resemble the in vivo native cardiac tissues (Figure 1).

Initial pioneering works showed that 3D reconstitution of embryonic chick [159] or neonatal rat [40] CMs in a collagen matrix generated the rod-type [159] or ring-shaped [40] EHTs that displayed more physiological and morphologically organized characteristics of intact differentiated heart tissues than monolayer cultures, serving as promising materials for in vitro assessments of cardiac function. Afterwards, the same and other groups further developed engineered cardiac microtissues on synthetic polymer polydimethylsiloxane (PDMS) racks [41] or on fluorescent pillars [160], highlighting the potential usefulness of these EHT models for various applications, such as in vitro drug screening and disease modeling. Another simple way of microfabrication is to pattern material substrates with biomimetic approaches, so that the cultured CMs are well organized and aligned. Feinberg et al. used PDMS thin films and assembled biohybrid materials by culturing neonatal rat CMs on the films that were micropatterned with alternating 20-μm-wide lines of a large amount of ECM proteins (fibronectin) [148]. This hybrid centimeter-scale construct improved the CM alignment and performed diverse biomimetic cardiac functions with fine spatial and temporal control, which served as a model for investigating the biomechanics and physiology of in vitro myocardial sheets. Ma et al. established more advanced culture systems, i.e., self-organizing human cardiac microchambers that were regulated by geometric cellular confinement [142]. They adopted PEG- and PDMS-patterned substrates to geometrically confine the hPSC colonies in circular chambers (200,400, and 600 μm in diameter), and provided cells with the biochemical and biophysical cues to induce self-organizing cardiac lineage specification and the creation of a beating human cardiac microchamber, which could be used to study early cardiac development and drug-induced developmental toxicity. Another good example of patterning culture chambers is the “Biowire^TM^” platforms, reported by Radisic’s group [45,151,152]. They combined 3D cell cultivation of hPSC-derived CMs and fibroblasts in hydrogel-coated microwells containing two flexible wires that were fabricated from a poly(octamethylene maleate (anhydride) citrate) (POMaC) polymer with electrical stimulation to mature hPSC-derived cardiac tissues. They demonstrated that the engineered platform markedly induced CM maturation, such as increased myofibril ultrastructural organization, elevated conduction velocity, and improved both electrophysiological and calcium handling properties when the cells were subjected to high-frequency electrical stimulation, compared to the non-stimulated or low frequency-stimulated controls. This platform has been further shown to be a suitable tool for studying in vitro drug testing and disease modeling of cardiac tissues [152].

With a sophisticated approach using engineered epicardial mimetics, Bian et al. developed a large 3D cardiac patch (2.5 cm × 2.5 cm) with human epicardial fiber orientations that were derived from diffusion tensor magnetic resonance imaging (DTMRI) maps of the human ventricle [140,150]. After three weeks in the culture of rat neonatal CMs on a PDMS mold with hexagonal posts corresponding to the designed porous photomask, they obtained the advanced structural and functional maturation of the engineered 3D cardiac tissues with CM alignment that replicated human epicardial fiber orientations. The same group also developed another versatile EHT, termed “cardiobundles”, on a PDMS frame, which was fabricated on dynamic, free-floating culture conditions with neonatal rat- or hPSC-CMs embedded in a fibrin-based hydrogel [43]. Notably, the cardiobundles showed a near-adult functional and structural characteristics without loading exogenous electrical or mechanical stimulation in vitro. Recently, the advanced microelectronic cardiac patch that integrated neonatal rat CMs with flexible, freestanding electronics and a 3D nanocomposite scaffold has been reported [147]. In the system, the electronic mesh device was designed to contain electrodes for the sensing of tissue electrical activities, which enabled it to provide online monitoring of tissue physiological functions, to stimulate cells and tissues electrically, and to also regulate the release of biomolecules in accordance with the tissue microenvironment. Thus, this complex cardiac patch that was integrated with microelectronics holds great promise for therapeutic monitoring and regulation of cardiac function on demand.

One of the key issues in cardiac tissue engineering is how to acquire and maintain the supply of oxygen and nutrients into the thick cardiac muscles. Recent development to resolve this issue is utilizing perfusable 3D microfluidics/microchannel devices, which are fabricated with human ECs and form the biomimetic microvasculature, functioning as engineered heart- or vasculature-on-a-chip [149,157]. The AngioChip platform has been shown to support the assembly of parenchymal cells on a mechanically-tunable matrix surrounding a perfusable and branched microchannel network, and to allow functional vascularization and perfusion through a maintained open-vessel lumen, generating both in vitro cardiac and hepatic tissue models with defined vasculature and in vivo cardiac tissue implants with direct surgical anastomosis in rat hindlimbs [149]. During the last decade, human heart organoids have attracted much attention as novel promising EHT models, although progress in this arena has been hampered and delayed significantly, compared to other organ organoid models [161,162,163]. Recent attempts have tackled this issue of the shortage of relevant heart-on-a-chip and/or heart organoid models, and thereafter, self-assembling organoid technologies have been established and become available for the heart [151,164,165,166]. For instance, Drakhlis et al. have reported hPSC-derived hybrid cardiac-foregut organoids, which resembled early native heart development before heart tube formation through an interaction with foregut endoderm tissues [143]. Interestingly, Hofbauer et al. have reported the self-organizing “cardioids” from hPSCs, which built chamber-like structures containing a cavity with isolated CM and EC layers and mimicked epicardial spreading, inward migration, and differentiation [144]. For a more in-depth review about the topics in regards to organ (heart)-on-a-chip and/or heart organoid models, we direct the readers to other reviews on these topics [151,164,165,166,167,168].

### 4.2. Bioprinting

The basic 3D scaffold fabrication techniques, such as hydrogels and decellularization, have been broadly used for tissue engineering of various organs, however, the construction of highly vascularized and dynamic organs such as heart is still challenging by using such conventional methodologies. Alternatively, 3D bioprinting is an emerging fabrication technique that creates biomimetic products with precise control over the positioning of living cells, biomaterials, and biomolecules, collectively termed bioinks [169,170]. In bioprinting of cardiac tissues, synthetic and/or natural materials, including collagen, gelatin, fibrin, alginate, hyaluronic acid, PCL, PLGA, and decellularized ECM scaffolds, are processed together with cardiac cells and/or biomolecules to pattern and recreate heart tissues precisely at high resolution (micro ~ nanoscale) with computer-aided designing modalities such as computer tomography and magnetic resonance imaging [68,154,171,172,173]. Based on the used technology, cost, speed, and resolution, there are different types to conduct the 3D bioprinting processes, namely inkjet, extrusion, laser-assisted, stereolithography, and droplet bioprinting, among which the extrusion bioprinting is the most frequently used approach to print cell-laden hydrogel-mixed cardiac tissues directly through a layer-by-layer deposition technique [68,169,171,174].

Using the extrusion-based multi-material bioprinting with flexible thermoplastic polyurethane (TPU) and conductive carbon black nanoparticles, Lind et al. manufactured a cardiac microphysiological device, in which soft strain gauge sensors were integrated within TPU microfilaments that guided the self-assembly and alignment of laminar cardiac tissues, providing non-invasive, electronic measurement of tissue contraction and an efficient platform to study drug responses [153]. Feinberg’s group developed an extrusion-based freeform printing methodology, termed “FRESH” (i.e., Freeform Reversible Embedding of Suspended Hydrogels), to bioprint collagen for engineering components of the human heart from multiscale vasculature and tri-leaflet heart valve to acellular neonatal-scale human heart (37 mm in diameter, 55 mm in height) [154,155]. In their system, controlling pH-driven gelation of collagen provided up to 10 μm resolution on printing, and hESC-CMs with cardiac fibroblasts (in a 49:1 ratio) could be embedded in the collagen and cellular bioinks. Importantly, using this approach, the authors also demonstrated successful 3D printing of a contractile human cardiac ventricle model (an ellipsoidal shell-shape; 6.6 mm in diameter, 8 mm in height). On the other hand, Gao et al. advanced a novel multiphoton-excited 3D printing technique to produce a native-like ECM scaffold with submicron resolution, in which multiple cell types including hiPSC-derived CMs, SMCs, and ECs (in a 2:1:1 ratio) could be seeded to generate a human cardiac muscle patch [79]. Notably, the engineered cardiac patch began producing calcium transients and beating synchronously within one day after seeding, and in vitro electrophysiological and contractile parameters of the patch were significantly developed and matured over the next seven days. The authors also showed the regenerative therapeutic potential of the engineered cardiac patch with successful cell engraftment when transplanted into damaged hearts in a murine MI model.

One of the advantages in the biofabrication by bioprinting is the high feasibility to create perfusable vasculature that is essential for successfully maintaining the structure and function of the engineered tissues and organs. Using a novel hybrid strategy with extrusion-based bioprinting, Zhang et al. engineered one of the first endothelialized myocardium that was capable of spontaneous and synchronous contraction through sequential steps, including bioprinting of a microfibrous scaffold encapsulating human ECs, formation of the vascular bed through EC migration, and seeding of human CMs onto a layer of endothelium [157]. The authors further embedded the organoids into a microfluidic perfusion device to successfully produce the heart (endothelialized myocardium)-on-a-chip platform for the purpose of testing cardiovascular toxicity. On the other hand, using a laser/visible light-assisted stereolithography technique that enables higher resolution bioprinting than extrusion-based methods, Grigoryan et al. manufactured high-fidelity multivascular networks within photopolymerizable hydrogels by using a new set of biocompatible photo-absorbers [156]. Although not in cardiac tissue engineering, they further showed the potential translational utility of their 3D-printed capillary vascular networks both in an engineered alveolar model and in a murine liver injury model. Another new biomanufacturing method that relied on sacrificial writing into functional tissue (SWIFT) has recently been reported by Skylar-Scott et al. [158]. They first assembled hundreds of thousands of hiPSC-derived organ building blocks (OBBs) in the forms of embryoid bodies, organoids, or multicellular spheroids, which were then placed into a mold and compacted via centrifugation to form a living OBB matrix, followed by quick patterning of a sacrificial ink within this matrix via embedded 3D bioprinting and induction of perfusable vascular channels after ink evacuation. As an interesting example, the authors created a perfusable cardiac tissue (top width, 6 mm; bottom width, 4.2 mm; depth, 4.2 mm; and height, 12 mm) that fused and beat synchronously over a seven-day period. More recently, Kupfer et al. have developed a photo-crosslinkable bioink formulation that is composed of ECM proteins and hiPSCs (instead of CMs), which were 3D-printed into complex geometries with a perfusable structure and, thereafter, enabled robust expansion and subsequent in situ differentiation of hiPSCs into electromechanically functional and chambered cardiac muscle organoids [175]. This method provides an advance in terms of in situ hPSC proliferation and differentiation at high density and a platform for achieving thick layers of contiguous cardiac muscles.

Overall, although fully functional whole organ (heart) reconstruction is not yet feasible, the bioprinting technologies are rapidly developing and continuously hold great promise now and in the future. However, several obstacles and limitations need to be addressed for further advancing this technology. First, similarly to hydrogels, there are issues in relation to foreign body reaction and arrhythmias upon transplantation of biofabricated/bioprinted constructs, which needs to be fully elucidated to ensure their functional and safe engraftment [176,177,178]. Second, vascularization as well as innervation of engineered/bioprinted tissues will be the key milestones for the generation of functionally engrafted constructs, however, only limited progress has been made in the design of innervation to date [179,180]. The development of an ideal bioink is essential to enhance the cell-cell and cell-biomaterial/environment interactions for obtaining increased sustainability of the bioprinted constructs. Printing speed also needs to be improved, since additive printing with a high level of detail requires a slow printing velocity, affecting the scalability of the current methods [181]. Other issues around the use of bioprinted constructs include the lack of standardized methods across these systems, the need of further improved image acquisition with high resolution, and costs. For a more in-depth review about bioprinting in cardiac tissue engineering, we direct the readers to other reviews [68,171,182,183].

## 5. Clinical Applications of Cardiac Tissue Engineering-Based Technologies

### 5.1. Drug Screening and Disease Modeling

The current hPSC and cardiac tissue engineering technologies can provide in vitro sophisticated 3D culture models of the human heart, which is more advanced resembling the native heart compared to the conventional 2D monolayer culture system. Therefore, in recent years, applications of these technologies have been widely expanded into generating various disease settings, in order to test drug effects to cardiac function and toxicity preclinically and to model cardiovascular diseases in combination with patient-derived hiPSCs.

Drug screening using hPSC-CMs is conducted to predict human cardiac responses against drugs and determine the potential functions of the drugs, such as proarrhythmic, (positive and negative) inotropic, and cardiotoxic effects [42,152,184,185]. For drug testing, animal models, primarily rodents, have been commonly used so far; however, that platform often failed to precisely predict the human drug responses due to differences in cardiac physiology and biology between rodents and humans (e.g., differential roles of the rapid delayed rectifier I_Kr_, such as “no role” in rodents vs. “strongly functioning” in humans) [186,187]. Therefore, hPSC-derived EHTs or similar 3D constructs, such as spheroids/organoids, microtissues, and microfluidic devices (“heart-on-a-chip”), have been emerging as better solutions for more precise drug screening in the heart, overcoming the deficits in the traditional 2D in vitro culture platforms [51,76,160,188]. In fact, using these advanced 3D culture systems, several higher predictive insights have been shown with regard to inotropic effects (e.g., isoprenaline) [189,190,191], contractile function [185], conduction disturbance [42,152], I_Kr_ blockade [77], and multifaceted proarrhythmic effects [41]. Despite these promising outcomes, the cardiac tissue engineering-based drug screening approach is still in its infancy and needs to be further improved for their reliability, reproducibility, and robustness. For instance, considering the larger costs and time-consuming efforts with the specific and expensive equipment that is required in the 3D EHT culture platforms, direct comparisons of drug screening between 2D and the various 3D systems are necessary to obtain insights and fair judgements about the benefits, demerits, and cost-effectiveness in both methods [6,192]. Another concern is that the 3D EHT models are not appropriate for high-throughput drug screening, since their complexity and bigger size than the 2D system likely make it incompatible for rapid screening of compounds in large libraries [160,193,194].

The 3D EHT models combined with hPSC-CMs have also advanced cardiac disease modeling to understand disease-specific phenotypes. In addition to the use of patient-derived hiPSCs with known disease-causing gene mutations, the recent gene editing technology such as CRISPR-Cas9 has enabled the quick generation of genetically-modified hPSCs (wild-type hESC- or hiPSC-derived) harboring a specific mutation, which can be also applied to disease modeling [195,196]. Importantly, this technology can correct a disease-causing gene mutation in patient-derived cells and thereby generate an isogenic control that provides validity and robustness of the obtained findings [197,198]. The hPSC-CM and 3D EHT-combined platforms harboring disease-causing genetic mutations, including hypertrophic cardiomyopathy (HCM) [199], dilated cardiomyopathy (DCM) [200], and Barth syndrome with mitochondrial cardiomyopathy [198] have successfully modeled in vitro disease phenotypes such as abnormal contractile properties, which were in accordance with each of the clinical phenotypes. It is crucial to consider that these disease-carrying hPSC-CMs, as well as mutation-corrected isogenic control hPSC-CMs, that are used in the disease-modeling platforms are still immature and thus may not reflect the characteristics in adult diseased CMs. In fact, hiPSC-CMs harboring a mutation truncating the sarcomere protein titin (*TTN*), a known common cause of DCM, did not show any abnormal contractile function and were comparable with wild-type hiPSC-CMs in a 2D culture platform but exhibited significantly reduced contractility and decreased sarcomere length when fabricated in 3D constructs of self-assembling cardiac microtissues that worked against the elastic resistance between two PDMS pillars [200]. Another insightful example to apply the findings that were obtained from disease modeling with the hPSC-CM and 3D EHT-combined platforms is their clinical translation for patient treatment in a way of personalized medicine. More recently, Prondzynski et al. have reported that HCM patient (harboring a rare mutation in the α-actinin 2 [*ACTN2*] gene)-derived hiPSC-CMs and engineered 3D muscle strips recapitulated several hallmarks of HCM, such as myofibrillar disarray, hypercontractility, impaired relaxation, and prolonged action potential duration, and that an _L_-type calcium channel blocker diltiazem improved these disease-specific cardiac abnormalities more significantly in the HCM models than in an isogenic control in their systems (Figure 2) [201]. The authors further translated their findings to patient care and showed that diltiazem application ameliorated the prolonged QTc interval in the HCM-affected son and sister of the index patient after one month treatment, providing a proof of concept for the use of the hPSC-CM and 3D EHT-combined platforms for personalized treatment of cardiomyopathies.

These personalized approaches are promising but at the same time, require a great amount of efforts and elevated costs. Thus, it is important to verify whether the larger costs and efforts in a personalized treatment approach compared to conventional approaches are outweighed by better sensitivity, specificity, robustness, and thereby outcomes.

### 5.2. Heart Regenerative Therapy

One of the most promising applications in cardiac tissue engineering is to use the technologies for therapeutic purposes such as regeneration and repair of the diseased hearts. Firstly, cell-based therapies for heart regeneration using i.e., bone marrow-derived mononuclear cells, mesenchymal stem cells, cardiac progenitor cells, or hPSC-CMs, have been developed and applied in the preclinical and clinical studies during the last two to three decades [9,203,204,205,206]; however, the therapeutic effects of these approaches are still undetermined and under debate. For a more in-depth review about cell therapies for heart regeneration, including the latest hPSC-CM strategies [30,31,207,208], we direct the readers to other reviews [5,22,209,210,211].

The main common bottleneck in cell-based approaches for heart regeneration is the low engraftment and survival rate of the transplanted cells in the host heart, which results in the fact that the transplanted cells cannot engraft into or replace the damaged cardiac tissues in a functionally sufficient manner [7,8,9,10,212]. In this regard, the recent advanced cardiac tissue engineering technologies are generating a lot of attention since they have been shown to attenuate pathological ventricular remodeling after heart injury, and when combined with cell therapies, to promote engraftment of the transplanted cells in the host tissue, leading to improvement of the LV function of the damaged hearts [6,11,12,23]. The main devices that are used for regenerative tissue-engineering purposes to date are hydrogel/ECM-based constructs (epicardial patches, or in situ gelling systems via intramyocardial or intracoronary injections), engineered muscle strips (applied on epicardium), and cell sheets (applied on epicardium) [81,132,137,213,214,215]. Related cardiac regenerative bioengineering studies in the preclinical settings using animal models are already described in the above sections and also highlighted in Table 2. Overall, these regenerative tissue-engineering approaches have showed the beneficial outcomes, including higher percentages of cell retention and improved ventricular morphology and function, when applied into infarcted or defected hearts in small and large animal models [82,97,139,213,214,215,216], and therefore, hold great promise for future heart regenerative medicine. However, several important issues, involving induction of arrhythmias due to inadequate electric coupling to the host myocardium, potential tumorigenicity when combined with hPSC-based cell therapies, long-term safety, adverse host immune responses attacking the cells and/or materials that are employed in the bioengineered constructs, and the side effects that are derived from the accompanied long-term immunosuppressive therapies, need to be addressed for further applying these technologies to the clinical settings [208,217,218,219,220].

Nonetheless, after obtaining promising results in preclinical settings, several cardiac bioengineering-based therapies have been and are being tested for their safety, feasibility, and (preliminary) efficacy in clinical settings (Table 2). The first-in-man clinically prospective randomized trials evaluating the therapeutic effects of acellular alginate-based hydrogels to the damaged hearts in patients with ischemic or nonischemic HF were PRESERVATION I [221,222,223] and AUGMENT-HF [224,225,226]. In these trials, the alginate-based hydrogels were administered via intracoronary (PRESERVATION I) or intramyocardial (AUGMENT-HF) injections in patients with acute ST-segment elevation MI two to five days after successful primary percutaneous coronary intervention (PRESERVATION I) or in patients with advanced chronic HF (57% ischemic and 43% nonischemic; AUGMENT-HF). Despite prior expectations, these approaches could not attenuate pathological LV remodeling or improve the LV function with only minor improvements in symptoms and clinical status, compared to the control [223,226], suggesting that a cell-free strategy may possibly have limitations and be insufficient to effectively regenerate and repair damaged hearts in humans. Focusing on the restoration of ECM homeostasis and structure in the heart environment, other clinical trials evaluating the therapeutic effects of ECM epicardial patches (CorMatrix^TM^-ECM; porcine small intestinal submucosa-derived) [227,228,229] or cardiac ECM hydrogels (VentriGel; decellularized porcine myocardium-derived) [230] to the recent or late infarcted hearts have recently been completed. The former was delivered on epicardium in patients that were undergoing CABG surgery within six weeks after acute MI as an adjunct therapy to surgical revascularization. The latter was delivered via intramyocardial (transendocardial) injections using the NOGA-MyoStar catheter system [234] in patients with LV dysfunction, who had experienced a first, large ST-segment elevation MI that was treated by percutaneous coronary intervention in the past 60-day to three-year window. The results for these studies have not yet been published to date, and therefore are awaited.

The first clinical application of cardiac bioengineered constructs containing hPSC-derived cells, namely clinical-grade hESC-derived CD15 (SSEA-1)^+^ISL-1^+^ cardiac progenitors that were embedded into a fibrin patch, has been employed during concomitant CABG or mitral valve surgery for the treatment of patients with severe ischemic HF (LV EF ≤ 35%) due to a history of MI (older than six months) [231,232]. The transplantation procedures of the progenitor cell-fibrin hybrid epicardial patches were all safely conducted in six patients with no adverse complications in the follow-up periods (a median of 18 months), i.e., no detection of tumors such as teratomas, no occurrence of arrhythmias, and no major alloimmunization events even if the immunosuppressive therapy was ceased one month after the surgery [232]. Although no significantly beneficial results with regard to efficacy such as LV EF were achieved, the symptoms in all of the hybrid patch-transplanted patients were improved along with an increased systolic motion of the treated heart segments, indicating that this cell-fibrin hybrid patch would be a promising strategy as a novel alternative therapy for ischemic heart disease. Other active and recently recruiting clinical trials will evaluate the efficacies of hiPSC-derived engineered heart muscle (EHM) for advanced HF (LV EF ≤ 35%) (NCT04396899) and hiPSC-CM sheets for ischemic cardiomyopathy (NCT04696328). The EHM in the former trial is constructed from hiPSC-derived CMs and stromal cells that are embedded into a bovine collagen type I hydrogel, aiming the myocardial remuscularization and the improvement of ventricular wall motion upon transplantation. The latter trial is based on promising results in the preclinical studies showing that allogenic transplantation of induced pluripotent stem cell-derived CM sheets improved cardiac function and remodeling with a concomitant immunosuppressive therapy in non-human primate MI models [217,233].

Collectively, the positive outcomes that were obtained in these clinical trials have been marginal or elusive to date, and therefore, development of the cardiac tissue engineering-based regenerative therapies is still in its infancy. Further advanced technologies and a better understanding of the underlying mechanisms through which these applied biomaterials and tools can elicit their cardiac regenerative properties would pave the way towards more improved bioengineering therapies for heart regeneration and repair.

## 6. Future Perspectives of Cardiac Tissue Engineering

Cardiac tissue engineering is a remarkably developing research field year by year, in particular during the last decade, enabling the generation of constructs that resemble the human native heart. As described above, it holds great potential for applications in drug screening, disease modeling, and heart regenerative therapies towards personalized medicine. For instance, using the combined technologies of advanced 3D bioprinting with hybrid material bioinks, high-resolution cardiac imaging modalities, and developed computational image processing, several universities and hospitals have already demonstrated patient-specific models of various cardiovascular diseases and pathologies [235,236]. Although such personalized bioengineering technologies and medicine are expected to become reality in the near future, continuous efforts are needed to assure the accuracy, relevance, reproducibility, and robustness in these engineered technologies and constructs. Since the cardiac tissue engineering field holds interdisciplinary aspects, the clinical translation and medical application of the engineered heart tissues will require tight collaborations across many disciplines, involving stem cell biology, material science, biofabrication technology, cardiovascular medicine, and also business administration [67,68,237].

One of the keys for generation and engraftment of the engineered functional cardiac tissues is to create perfusable vasculature that is accompanied with heart muscles. Scaling up the engineered constructs to human-sized tissues is also an essential milestone for success in cardiac bioengineering medicine. For efficient vascularization and scalability, as well as the modulation of the host immune response and attenuation of pathological ventricular remodeling, developing well-controlled biomolecule release systems via biomaterials is likely to provide a promising solution. For those purposes, using the novel delivery systems with e.g., chemically-modified mRNA [238,239,240] or exosome such as extracellular vesicles [241,242,243] may offer an alternative option, which will need to be proven by further investigation.

In conclusion, the state-of-the-art cardiac tissue engineering technologies hold unlimited potential and their broad applications to drug screening, cardiovascular disease modeling, and heart regenerative medicine. Although the clinical-grade effects of the technologies to remuscularize failing hearts and to improve the cardiac function and survival in patients with severe HF are still uncertain, the advanced cardiac bioengineering technologies definitely not only provide novel insights in cardiac biology and medicine but will open new therapeutic paths of regeneration and repair for heart disease on the horizon.

## Figures and Tables

**Figure 1 ijms-23-03482-f001:**
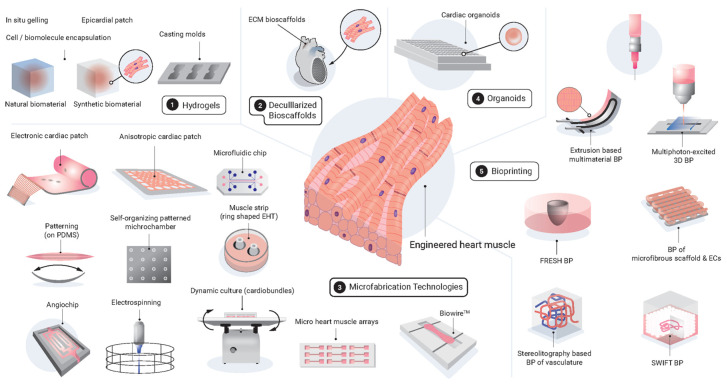
**Schema of the latest biomaterials and biofabrication technologies for the generation of engineered heart muscles.** 3D, three-dimensional; BP, bioprinting; EC, endothelial cell; ECM, extracellular matrix; EHT, engineered heart tissue; FRESH, freeform reversible embedding of suspended hydrogels; PDMS, polydimethylsiloxane; SWIFT, sacrificial writing into functional tissue. **1. Hydrogels:** In situ gelling [82,97], Epicardial patch [83,95], Cell/biomolecule encapsulation [64,75], Casting molds [40,76]; **2. Deculllarized Bioscaffolds**: ECM bioscaffolds [134,135]; **3. Microfabrication Technologies:** Electronic cardiac patch [147], Patterning (on PDMS) [148], Angiochip [149], Anisotropic cardiac patch [140,150], Self-organizing patterned michrochamber [142], Electrospinning [145,146], Microfluidic chip [151], Muscle strip (ring shaped EHT) [40], Dynamic culture (cardiobundles) [137], Micro heart muscle arrays [76], Biowire^TM^ [45,152]; **4. Organoids**: Cardiac organoids [143,144]; **5. Bioprinting:** Extrusion based multimaterial BP [153], FRESH BP [154,155], Stereolitography based BP of vasculature [156], Multiphoton-excited 3D BP [79], BP of microfibrous scaffold & ECs [157], SWIFT BP [158].

**Figure 2 ijms-23-03482-f002:**
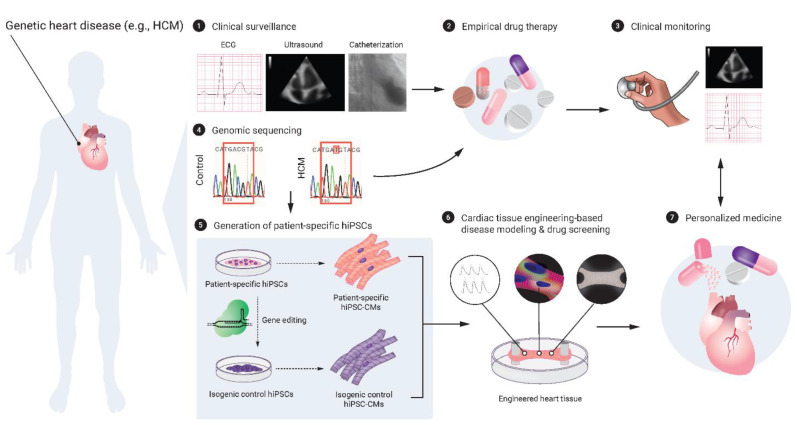
**Modeling of cardiac tissue engineering-based personalized medicine for genetic heart disease.** The panel highlights the hPSC-CM and 3D engineered heart tissue-combined platforms for the personalized treatment to genetic heart disease, as Prondzynski et al. have demonstrated such a proof-of-concept successfully to the inherited HCM-affected family, harboring a rare c.740C>T mutation in the *ACTN2* gene [201]. The image of catheterization was obtained from a previous report [202]. CM, cardiomyocyte; HCM, hypertrophic cardiomyopathy; hiPSC, human inducible pluripotent stem cell.

**Table 1 ijms-23-03482-t001:** **Tissue-engineering strategies for promoting maturation in hPSC-CMs.** 3D, three-dimensional; EC, endothelial cell; EHT, engineered heart tissue; FB, fibroblast; SMC, smooth muscle cell.

Major Classification	Type of Strategies	Subtype	References
Physiological stimulation	Mechanical stress loading		[40,41,42,43]
	Electrical stimulation/pacing		[42,44,45]
Advanced culture	3D cultures	EHTs with a scaffold protein	[46,47,48,49]
		Cardiac spheroids without scaffold proteins	[47,50,51]
	Intercellular crosstalk	Co-culture/administration with ECs	[52,53]
		Co-culture/administration with FBs	[48]
		Co-culture/administration with epicardial cells	[54]
		Co-culture/administration with SMCs/ECs and FBs	[55,56,57,58]
Biochemical intervention	Metabolic intervention	High fatty acid/low glucose diet	[59,60]
	Hormone intervention	Tri-iodothyronine (T3)	[61,62]
		Glucocorticoid	[63]
	Paracrine signals	Growth factors	[23,64]
Genetic regulation	Overexpression of maturation-inducing genes	Ion channels (e.g., *KCNJ2*), etc.	[23,32]
	Induction of cell cycle arrest		[23,32]
In vivo transplantation	Transplantation into a normal or injured heart		[32,65]

**Table 2 ijms-23-03482-t002:** **The selected in vivo preclinical studies and human clinical trials of cardiac tissue engineering-based therapies for heart regeneration and repair.** CABG, coronary artery bypass grafting; CM, cardiomyocyte; EC, endothelial cell; ECM, extracellular matrix; EF, ejection fraction; EHM, engineered heart muscle; hESC, human embryonic stem cell; hiPSC, human inducible pluripotent stem cell; HF, heart failure; IGF, insulin-like growth factor; IJ, injection; IR, ischemia reperfusion injury; MI, myocardial infarction; PLGA, poly(lactic-*co*-glycolic acid); PNIPAAm, poly(N-isopropylacrylamide); SMC, smooth muscle cell.

** In Vivo Preclinical Studies **							
**Tissue Engineering Strategy**	**Biomaterial**	**Cell Source**	**Animal**	**Disease Model**		**Route of Administration**	**References**
Cardiac patch	Fibrin	hiPSC-CM, pericytes	Rat	MI		Epicardium	[95]
Cardiac patch	Fibrin (IGF-loaded)	hiPSC-CM, -EC, and -SMC	Pig	MI		Epicardium	[97]
Muscle strip/EHM	Fibrin	hiPSC-CM, -EC	Guinea pig	Cryoinjury		Epicardium	[139,213]
EHM	Collagen	hESC-CM	Rat	MI/IR		Epicardium	[99]
EHM	Collagen	hESC-CM	Rat	Chronic MI/IR		Epicardium	[81]
In situ gelling system	Alginate	Acellular	Rat/Pig	MI		Intramyocardial/ Intracoronary IJ	[102,103]
In situ gelling system	Chitosan	Acellular	Rat	MI		Intramyocardial IJ	[106]
Nanofibers/EHM	PLGA	hiPSC-CM	Rat	MI		Epicardium	[110]
In situ gelling system	PNIPAAm-based	Acellular	Rat	Chronic MI		Intramyocardial IJ	[112]
In situ gelling system	Decellularized cardiac ECM hydrogel	Acellular	Pig	MI		Trans-endocardial IJ	[132]
3D-printing-based cardiac patch	ECM scaffold	hiPSC-CM, -EC, and -SMC	Mouse	MI		Epicardium	[79]
Cell sheet	Scaffold-free	hiPSC-CM	Pig	MI		Epicardium	[137]
** Human Clinical Trials **							
**Trial Name** **(Trial Identifier)**	**Patient Population**	**Phase**	**Tissue Engineering Strategy**	**Biomaterial**	**Cell Source**	**Route of Administration**	**References**
PRESERVATION I (NCT01226563)	Acute MI; Congestive HF; ST elevation MI	Phase 1 completed	In situ gelling system	Alginate-based hydrogel	acellular	Intracoronary IJ	[221,222,223]
AUGMENT-HF (NCT01311791)	Severe chronic HF (EF ≤ 35%) [57% ischemic and 43% nonischemic]	Phase 2 completed	In situ gelling system	Alginate-based hydrogel	acellular	Intramyocardial IJ	[224,225,226]
Epicardial infarct repair using CorMatrix^TM^-ECM (NCT02887768)	Acute~subacute MI (within 6 weeks); CABG scheduled	Phase 1 completed	Cardiac patch	Porcine small intestine-derived ECM epicardial patch	acellular	Epicardium	[227,228,229]
VentriGel in post-MI patients (NCT02305602)	Recent ~ late MI (60 days to 3 yrs); Ischemic HF (EF ≥ 25%; ≤45%)	Phase 1 completed	In situ gelling system	Porcine myocardium-derived ECM hydrogel	acellular	Trans-endocardial IJ	[230]
ESCORT (NCT02057900)	Severe chronic ischemic HF (EF ≤3 5%)	Phase 1 completed	Cardiac patch	Fibrin patch	hESC-derived CD15^+^Isl1^+^ progenitors	Epicardium	[231,232]
BioVAT-HF (NCT04396899)	Severe HF (EF ≤ 35%)	Phase 1 & 2 recruiting	EHM	Collagen type I hydrogel	hiPSC-CM, -stromal cell	Epicardium	
Human (allogenic) iPSC-CM sheet for ischemic cardiomyopathy (NCT04696328)	Severe chronic ischemic HF (EF ≤ 35%)	Phase 1 recruiting	Cell sheet	Scaffold-free cell sheet	hiPSC-CM	Epicardium	[217,233]

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
