# Peer review of "Progress in Bioengineering Strategies for Heart Regenerative Medicine"

_ijms, 2022, doi:10.3390/ijms23073482_

Round 1
Reviewer 1 Report
In this narrative review the Authors summarize and discuss the currently available bioengineering approaches for heart regeneration. The review is well organized and well written. However, the manuscript is a bit too descriptive in some parts, and it fails to put the cited original articles in a “bigger picture”, highlighting their potential as well as their limitations, keeping in mind the potential impact in improving patients’ life.
More specifically, limitations of the described bioengineering strategies are not properly discussed (with the exceptions of the Recellularization Strategies and Heart Regenerative Therapies). Although this reviewer generally agrees with the positive approach taken by the authors, the discussion of the current limitations in hydrogel use (e.g., foreign body reaction and the need for engineering the gel towards tissue homeostasis, etc.), bioprinting, (e.g., vascularization and gas and nutrient exchange, limited of cell-cell and cell-enviroment interaction present in several of the bioprinting works discussed). Moreover, a discussion on the clinical bottlenecks to clinical implementation of some citated approaches (e.g., elevated costs in for several of the personalized approaches) is missing, This is well presented in the section 5, and a similar work should be don in the other sections.
Minor Points:
- Line 28-30, the sentence needs a small revision.
- Lines 57-60, the sentence seems incomplete.
- Line 196, the sentence misses its initial part.
Author Response
We appreciate the time and consideration this reviewer took to review our manuscript in a stringent and fair manner. We are pleased to respond to his/her questions and concerns in a point-by-point fashion as below.
[Major point]
Thank you for the important point and suggestion. We have newly discussed the obstacles and limitations in these engineering technologies with appropriate references in session 3 "functional biomaterials" – 3.1. Hydrogels (Line 258-280), session 4 "implementation of biofabrication tools" – 4.2. Bioprinting (Line 479-492), and session 5 "clinical application" – 5.1. Drug Screening and Disease Modeling (Line 563-566).
[Minor points]
- Line 28-30: We have revised this sentence (Line 28-31).
- Line 57-60: We have revised this sentence (Line 57-60).
- Line 196: Due to some technical error, the words were missing. We have added the missing words (Line 195-196).
Finally, we again appreciate this reviewer’s valuable comments and suggestions that encourage us to further improve the quality of the manuscript.

Reviewer 2 Report
Haneke et al. make a good job to comprehensively summarize the recent advances of heart regeneration. However, there are some minor issues required to be clarified.
- Line 55: When we mention iPSCs, we use “induced” instead of “inducible”.
- Line 195 and 196: There are some missing words in the sentence.
- The study of Agisyl-LVR injection in CABG patients (reference 88) is a very small observational study, including only 3 patients. Therefore, we can not get any conclusion from this study. However, the authors cited this study to show the therapeutic benefits of agisyl-LVR to improve cardiac function in heart failure patients (Lines 200-207). This statement (paragraph) should be deleted.
- The authors comprehensively described the recent progress of “functional biomaterials” and “implementation of biofabrication tools” for cardiac tissue engineering. However, they should also mention the obstacles and weak points of these novel techniques.
- Session 5 “applications of cardiac tissue engineering for prevision medicine” is good. However, session 5.1 is not relevant to the topic and title of this manuscript.
- It seems that session 5.2 is not relevant to “precision medicine”. Please revise the title of session 5.2 or session 5.
- Please check the references. For example, reference 30 and reference 195 are the same.
Author Response
We appreciate the time and consideration this reviewer took to review our manuscript in a stringent and fair manner. We are pleased to respond to his/her questions and concerns in a point-by-point fashion as below.
- We have corrected that term (line 55).
- Due to some technical error, the words were missing. We have added the missing words (Line 195-196).
- Thank you for the input. As the large clinical trials with Algisyl-LVRTM are described in session 5.2, we have shortened and modified this part in session 3.1 (Line 203-206).
- Thank you for the important suggestion. We have newly discussed the obstacles and limitations in these engineering technologies with appropriate references in session 3 "functional biomaterials" (Line 258-280) and session 4 "implementation of biofabrication tools" (Line 479-492).
- We have revised the title of session 5 to "Clinical Applications of Cardiac Tissue Engineering-based Technologies" (Line 495).
- As noted in the response to Q5, we have deleted the term "Precision Medicine" in the title of session 5 (Line 495).
- Thank you for the point. We have deleted ref 195.
Finally, we again appreciate this reviewer’s valuable comments and suggestions that encourage us to further improve the quality of the manuscript.
